# Statistical Significance of Clustering for High-Dimensional Count Data

## Abstract

Clustering is widely used in biomedical research for meaningful subgroup identification. However, most existing clustering algorithms do not account for the statistical uncertainty of the resulting clusters and consequently may generate spurious clusters due to natural sampling variation. To address this problem, the Statistical Significance of Clustering (SigClust) method was developed to evaluate significance of clusters in high-dimensional data. While SigClust has been successful in testing mixtures of continuous distributions, it is not specifically designed for discrete distributions, such as count data in genomics. Moreover, SigClust and its variations often suffer from reduced statistical power when applied to non-Gaussian high-dimensional data. To overcome these limitations, we propose SigClust-DEV, a method designed to evaluate the significance of clusters in count data. Through extensive simulations, we compare SigClust-DEV against other existing SigClust approaches across various count distributions and demonstrate its superior performance. Furthermore, we apply our method SigClust-DEV to Hydra single-cell RNA sequencing (scRNA) data and electronic health records (EHRs) of cancer patients to identify meaningful latent cell types and patient subgroups, respectively.

## 1 Introduction

Clustering is a powerful unsupervised statistical tool, widely applied in biomedical research to better understand complex data. For instance, it has been used to annotate and discover cell types from scRNA data (Butler et al., 2018) and to identify cancer subtypes (TCGA et al., 2012). Specifically, when a clustering algorithm assigns a single known population to multiple clusters, it may indicate that the population is heterogeneous and contains underlying subgroups. However, an important question remains: Are the clustering results meaningful or pure artifacts of random sampling? Most current clustering workflows do not address this important issue.

In the literature, popular pipelines for clustering high-dimensional biomedical data typically follow these steps (Waltman & Van Eck, 2013): (1) Applying dimension reduction methods such as Principal Component Analysis (PCA) to the data, (2) Implementing clustering algorithms such as $k$-nearest neighbors, hierarchical clustering, or $k$-means (Macqueen, 1967) on the resulting principal components (PCs), and (3) Deciding the number of clusters by thresholding distances across clusters. The final number of clusters can also be determined through manual inspection of cluster stability or variance reduction (Peyvandipour et al., 2020; Tang et al., 2021). However, these approaches are prone to create spurious clusters even in simple settings. Taking $k$-means clustering as an example, the algorithm can separate data drawn from a one-dimensional Gaussian distribution to distinct clusters with large between-cluster distance. A two-sample $t$-test also gives significant $p$-value between clusters, suggesting that they are different from each other. However, in many applications, it is not desirable to divide data of a single normal distribution into several clusters, which may cause false discoveries of biomedical subtypes.

To address this challenge, Liu et al. (2008) proposed a Monte Carlo-based statistical significance method, SigClust, to evaluate clustering results in high-dimensional data. A key contribution of their work was the careful consideration of the definition of clusters. Specifically, SigClust adopts a probabilistic approach, assuming that multiple clusters represent a mixture of Gaussian distributions, while a single cluster that cannot be further divided approximates a Gaussian distribution. A

critical step in SigClust involves assessing the separation of clusters by treating the collected data as if it were drawn from a simple Gaussian distribution (i.e., the null distribution), using a Monte Carlo approach. To evaluate the separation of clusters, the SigClust uses cluster index (CI), the ratio of within-cluster variation over the total variation, as the test statistics. Then, SigClust compares the CI under the null distribution against that of the observed data to obtain a $p$-value. This allows for a formal statistical test to determine whether the underlying distribution of the data can be reasonably approximated by a single Gaussian distribution, i.e., one cluster. If the observed CI is not significantly different from the CI distribution under the null distribution, the SigClust concludes over-clustering and that the data should not be divided into subgroups. There are several extensions of the original SigClust. Huang et al. (2015) improved the estimation of the null distribution by introducing soft-thresholding of the covariance matrix estimation. Kimes et al. (2017) extended SigClust to hierarchical clustering, and Grabski et al. (2023) further adapted it for scRNA data. Most recently, Shen et al. (2024) generalized SigClust to clustering in the reduced multi-dimensional scaling (MDS) space.

Since the SigClust was introduced by Liu et al. (2008), it has been widely applied in various biomedical research problems, including cancer subtype identification (TCGA et al., 2012; Agrawal et al., 2014), cell type discovery (Boldog et al., 2018), and gene expression network analysis (Lee et al., 2021; Garcia-Recio et al., 2023) However, the significance of clustering for count data has not been thoroughly established. Recently, Grabski et al. (2023) adapted SigClust for scRNA counts by replacing the Gaussian null distribution in the original SigClust with the Poisson log-multivariate normal (log-MVN) distribution (Aitchison & Ho, 1989). Although this approach has successfully refined cell-type annotations, the presumed null distribution is difficult to estimate due to the high-dimensional nature of such data, leading to potential low power issues. Furthermore, this method is specifically designed for scRNA data and is not easily generalizable to other types of count data, such as binary data. Unlike the multivariate Gaussian distribution, the estimation of count data in high-dimensional spaces is often challenging, further complicating the original SigClust framework. Other methods for assessing clustering significance, including previous works of McShane et al. (2002), Maitra et al. (2012), Chakravarti et al. (2019), Chen & Witten (2023), and Gao et al. (2024) are not specifically designed for count data.

In this article, we propose a novel SigClust workflow for high-dimensional count data using the deviance-based PCA (DEV-PCA) space (Townes et al., 2019), namely SigClust-DEV. Deviance-based PCA is a powerful tool within generalized PCA approaches (Collins et al., 2002; Lee et al., 2010; Landgraf & Lee, 2020a;b), which are nonlinear dimension reduction methods that project data from the exponential family into the natural parameter space. The core idea of generalized PCA is to preserve the structure of heterogeneous natural parameters, which aligns well with the mixture definition of clusters used in SigClust. As mentioned earlier, current workflows for clustering count data rarely apply clustering algorithms directly on the discrete distribution space. Instead, they first project the count data into a latent space before performing clustering. Inspired by the work of Shen et al. (2024), which performed SigClust in the latent space from MDS, we extend SigClust to the latent space for count distributions. To improve the robustness of SigClust in latent spaces, we utilize the relative goodness of fit as the test statistics for SigClust-DEV (Chakravarti et al., 2019).

The rest of the paper is organized as follows. In Section 2, we introduce the related methods and describe the details of the proposed SigClust-DEV. Next, we investigate the performance of SigClust-DEV through comprehensive numerical experiments in Section 3. Then we apply SigClust-DEV to investigate two high-dimensional biomedical data: the scRNA data for *Hydra* stem cells (Siebert et al., 2019) and the medical records for cancer patients in Section 4. We conclude the article with some discussions in Section 5.

## 2 METHODOLOGY

In this section, we start with the description of SigClust and its variants in Section 2.1. Then we introduce the generalized PCA for exponential family in Section 2.2. Finally, SigClust-DEV is introduced in Section 2.3.

## 2.1 CLUSTER SIGNIFICANCE FOR MIXTURE OF GAUSSIAN DISTRIBUTIONS

The original SigClust (Liu et al., 2008) tests the null hypothesis that the data come from a unimodal distribution against a mixture of unimodal distributions i.e., $H_0 : P \sim P_0$ vs. $H_1 : P \sim \alpha P_1 + (1 - \alpha)P_2$. The test statistic of this problem is the $k$-means cluster index, defined as the ratio of the within-cluster variation to the overall variation,

$$CI = \frac{\sum_{a=1}^{k} \sum_{i \in C_a} \|\mathbf{x}_i - \bar{\mathbf{x}}^{(a)}\|_2^2}{\sum_{i=1}^{n} \|\mathbf{x}_i - \bar{\mathbf{x}}\|_2^2},$$

where $C_a$ is the index set of the $a$-th cluster, $\mathbf{x}_i$ denotes the $i$-th observation, and $\bar{\mathbf{x}}^{(a)}$ denotes the corresponding within-cluster mean. Under the alternative hypothesis, the observations are concentrated in each cluster and the the corresponding within-cluster variation tends to be small, leading to a small cluster index. Conversely, clustering on an unimodal distribution may result in small between-cluster variation and produce a relatively large cluster index.

Note that if $P_0$ is not specified, the null distribution of $CI$ is intractable. Therefore, SigClust assumes that $P_0$ is simply Gaussian, and then adopts a Monte Carlo procedure to iteratively generate $\hat{P}_0$ and estimate the empirical distribution of $CI$. Specifically, it first estimates the null distribution as $\mathcal{N}(\mathbf{0}, \hat{\Sigma}_n)$ , where the mean component is set to $\mathbf{0}$ since $CI$ is invariant across locations, and $\hat{\Sigma}_n$ is the estimated covariance matrix of the original data. Then it draws $n$ samples from the null distribution for $N_{sim}$ times. Finally, the empirical distribution of $CI$ can be estimated by applying $k$-means clustering on those generated null samples. The significance of clusters is assessed by the empirical $p$-value:

$$p = \frac{\#\{CI_m : CI_m \leq CI\}}{N_{sim}},$$

where $CI_m$ is the cluster index evaluated on the $m$-th batch of generated null samples. Interestingly, although the original SigClust exclusively considers the Gaussian mixtures, $CI$ measures the separation of clusters instead of the normality of the data. The Gaussian assumption is only used to approximate the null distribution of $CI$, and the $CI$ of a single Gaussian distribution is usually a robust yet conservative reference for many other continuous unimodal distributions, such as $t$-distribution and $\chi^2$-distribution (Shen et al., 2024).

A critical challenge of the original SigClust is that the null covariance $\hat{\Sigma}_n$ may not be accurately estimated under the high-dimensional settings. Although hard-thresholding and soft-thresholding has been used to improve the estimate of covariance matrix (Liu et al., 2008; Huang et al., 2015), SigClust can still suffer from low power due to the high dimensionality (Chakravarti et al., 2019). This may lead to conservative Type-I error and low power for SigClust. In contrast, MDS-based SigClust was proposed to avoid estimating the high-dimensional covariance matrix by projecting the original data into a low-dimensional space using MDS (Shen et al., 2024). Since the resulting MDS-space preserves the pairwise distance of the original data, the clustering structure can also be reserved (Abbe et al., 2022; Little et al., 2023). Therefore, after performing dimension reduction, results from both SigClust and distance-based clustering algorithms still align with those in the original data space. In practice, MDS finds a low-dimensional representation $\mathbf{Y}$ of a high-dimensional matrix $\mathbf{X}$ by minimizing the following reconstruction error:

$$\sigma_r(\mathbf{Y}) = \sum_{i,j}(d_{ij} - \delta_{ij})^2, \tag{1}$$

where $d_{ij} = d(\mathbf{x}_i, \mathbf{x}_j)$ and $\delta_{ij} = d(\mathbf{y}_i, \mathbf{y}_j)$. When the distance metric $d$ is the Euclidean distance, i.e., $d(\cdot) = \|\cdot\|_2$, MDS is equivalent to the standard PCA (Mead, 1992; Borg & Groenen, 2007). Although PCA does not explicitly require the original data to be Gaussian, its reconstruction error implicitly maximizes the multivariate Gaussian likelihood of $\mathbf{x}_i \sim \mathcal{N}(\boldsymbol{\mu} + \mathbf{V}\mathbf{u}_i, \sigma^2\mathbf{I}_{p \times p})$, where $\boldsymbol{\mu}$ denotes the mean vector, $\mathbf{V} = [\mathbf{v}_1, ..., \mathbf{v}_p]^T$ consists of an orthogonal basis in $\mathbb{R}^q$ called loadings, $\mathbf{u}_i$'s are the linear combinations of the loadings (i.e., principal components), defined as $\mathbf{u}_i = \mathbf{V}^T\mathbf{x}_i$, and $\sigma^2$ is the known variance. When the data are drawn from the Gaussian mixtures, the cluster structure can be recovered by learning the low-dimensional embedding $\mathbf{U} = [\mathbf{u}_1, ..., \mathbf{u}_n]^T$. However, when the data are drawn from some discrete distribution, the Euclidean MDS-space may fail to describe the cluster structure, resulting in undesirable clustering results.

## 2.2 Generalized PCA for Exponential Family

To obtain the representation of PCA for more general distributions, especially for count data, it is desirable to extend the standard PCA from the Gaussian distribution to the exponential family, analogous to the generalization of linear models to generalized linear models (GLM). Exponential family includes a large variaty of discrete distributions, including Binomial distribution, Poisson distribution, and Multinomial distribution, sufficient for modeling common count data for biomedical research.

Using the probabilistic interpretation of PCA, Collins et al. (2002) developed generalized PCA. Similar to the development of GLM, generalized PCA maximizes the likelihood of exponential family,

$$\min_{\{\theta_{ij}\},\phi} \prod_{i,j} \exp\left\{\phi[x_{ij}\theta_{ij} - b(\theta_{ij}) - c(x_{ij})] - \frac{1}{2}s(x_{ij},\phi)\right\}, \tag{2}$$

with constraint that requires the natural parameter to be embedded in a low-dimensional space, i.e., $\theta_{ij} = \mu_j + \mathbf{u}_i^T \mathbf{v}_j$. In other words, instead of preserving the pairwise Euclidean distance of the original data $\mathbf{X}$, generalized PCA approximates the canonical parameter matrix $\boldsymbol{\Theta} = [\theta_{ij}]_{n \times p}$ by the matrix $\mathbf{1}_n\boldsymbol{\mu}^T + \mathbf{U}\mathbf{V}^T$ from the linear subspace of $\mathbb{R}^q$, where $\mathbf{U}$ and $\mathbf{V}$ are $n \times q$ and $p \times q$ matrices. Returning to the problem of clustering, if the data follow a single count distribution in the exponential family, the natural parameter lies in a one-dimensional latent space, i.e., $\boldsymbol{\Theta} = \mathbf{1}_n\boldsymbol{\mu}^T$. Hence, the principal components $\mathbf{U}$ simply represent the noisy residuals of the natural parameters, which do not exhibit any cluster pattern by definition. In contrast, if the data follow a mixture of count distributions, the principal components $\mathbf{U}$ will learn the latent structure of the data and be well separated across clusters.

The optimization problem of generalized PCA (2) is usually challenging and may lead to unstable results (Lee et al., 2010; Townes et al., 2019; Landgraf & Lee, 2020b). Due to its computational issue, Townes et al. (2019) developed a two-step approximate algorithm, deviance PCA. Deviance PCA first partially optimizes $\boldsymbol{\mu}$ by fitting a GLM to each column of $\mathbf{X}$. Specifically, assuming that each column of data follows a distribution from the exponential family with unknown parameters $\{\theta_j, \phi\}$, this step computes the maximum likelihood estimators (MLEs) of those parameters and sets $\hat{\boldsymbol{\mu}} = (\hat{\theta}_1, ..., \hat{\theta}_p)^T$. Next, deviance PCA computes a deviance matrix $\mathbf{D}$ with elements defined by:

$$D_{ij} = \text{sign}(x_{ij} - \hat{\theta}_j)\sqrt{2\left[l(x_{ij},\hat{\phi}) - l(\hat{\theta}_j,\hat{\phi})\right]}, \tag{3}$$

where $l(\theta, \phi)$ denotes the log-likelihood for an entry of $\mathbf{X}$. This matrix approximates the canonical parameter matrix $\boldsymbol{\Theta}$ adjusted by the column-wise mean $\hat{\boldsymbol{\mu}}$. Finally, the principal components $\hat{\mathbf{U}}$ and the loadings $\hat{\mathbf{V}}$ are obtained by performing PCA on the deviance matrix. Notably, deviance-PCA can also be viewed as a version of MDS that preserves the pairwise distance of the deviance matrix $\mathbf{D}$.

## 2.3 Cluster Significance for Count Data

To assess the clustering signifiance for count data, an intuitive approach is to replace the assumption on $P_0$ from a single Gaussian distribution with a specific count distribution, such as Poisson distribution, and then replace the clustering algorithm from $k$-means to the generalized PCA-based clustering. For example, scSHC proposed by Grabski et al. (2023) assumes that the null distribution is a Poisson log-MVN distribution for scRNA data and uses generalized PCA for clustering. However, as mentioned above, estimation of the null distribution, especially for the covariance matrix, may be inaccurate due to the high-dimensionality. Moreover, the Poisson log-MVN model is too restrictive for scRNA data and not applicable for more general count distributions. In addition, parametrization of general high-dimensional count data is also challenging.

To address these challenges, we propose a novel approach to compute the significance of clusters for distributions from the exponential family, including Gaussian, Poisson, and Binomial distributions. Instead of directly estimating the multivariate count distribution as proposed by Grabski et al. (2023), we propose to first map the original data into a moderate dimensional latent space by generalized PCA (denoted as $\mathbf{Z}$), and then calculate the significance of clusters on the latent space using MDS-based SigClust (detailed in Appendix A, also denoting the data in the MDS-space as $\mathbf{Y}$).

Chakravarti et al. (2019) pointed out that the usage of $CI$ in the original SigClust only focuses on the null distribution. As a result, the test may not be very powerful in certain situations. Therefore, we utilize the relative goodness of fit as the test statistic. The basic idea is to test whether a mixture of Gaussian distributions better fits the data than a single Gaussian distribution. If the data are not Gaussian but with no clusters, the null distribution of $CI$ will be affected. In contrast, in terms of relative goodness of fit, a single Gaussian distribution can be a better fit than a mixture of Gaussian distributions when there are no clusters. Specifically, we compare the Kullback-Leibler (KL) divergence of the latent data distribution $P$ and a single Gaussian distribution $\hat{P}_0$ fitted on the data against a Gaussian mixture model $\hat{\alpha}\hat{P}_1 + (1 - \hat{\alpha})\hat{P}_2$ fitted on the data. The test is to see whether $H_0 : T := D_{KL}(P||\hat{P}_0) - D_{KL}(P||\hat{\alpha}\hat{P}_1 + (1 - \hat{\alpha})\hat{P}_2) \leq 0$. Under the null hypothesis, the latent data distribution can be better approximated by a single Gaussian distribution than a Gaussian mixture model, therefore $T < 0$. Conversely, under the alternative assumption, the latent distribution may be better approximated by a mixture of Gaussian distributions, hence $T > 0$. To estimate $T$, the following test statistics can be used:

$$\bar{T} = \frac{1}{n}\sum_{i=1}^{n}T_i := \frac{1}{n}\sum_{i=1}^{n}\log\left(\frac{\hat{\alpha}\hat{P}_1(Y_i) + (1 - \hat{\alpha})\hat{P}_2(Y_i)}{\hat{P}_0(Y_i)}\right), \qquad (4)$$

where $Y_i$ is the $i$-th observation in the low-dimensional latent space. The null distribution $\hat{P}_0$ is estimated using an overall Gaussian fit, and the alternative distributions $\hat{P}_1$ and $\hat{P}_2$ are fitted by the observations assigned to each cluster using separate Gaussian fits. Note that $\bar{T}$ is asymptotically normal conditioned on $\hat{\alpha}$, $\hat{P}_0$, $\hat{P}_1$, and $\hat{P}_2$. Therefore, in practice, we adopt a cross-fitting strategy by fitting the Gaussian mixtures on part of the observations and computing $\bar{T}$ on the rest of the data. In this way, a formal statistical test can be derived. Another approach is to use nonparametric tests, such as sign test, to evaluate if the median of $T_i$'s is larger than 0. Hence, the asymptotic normality of $\bar{T}$ is not required. Our proposed SigClust-DEV is summarized in Algorithm 1. In this paper, we use the nonparametric test to derive $p$-values. As a remark, we would like to point out that the relative goodness of fit is not suitable for high-dimensional data due to the need of estimating Gaussian mixture components. However, since SigClust-DEV works in the low-dimensional MDS space, relative goodness of fit can be performed effectively.

---

**Algorithm 1** SigClust-DEV

---

1. Set the dimension of generalized/deviance PCA space $s$ and the dimension of MDS space $t$.
2. Obtain the latent variables $\mathbf{Z}_{n\times s}$ by solving (2) or (3).
3. Obtain the MDS matrix $\mathbf{Y}_{n\times t}$ from the dissimilarity matrix $\mathbf{D}_{n\times n}$ of $\mathbf{Z}_{n\times s}$ by solving (1).
4. Randomly split the data into a training set $T$ and a validation set $V$.
5. Fit $\hat{P}_0$, $\hat{P}_1$ and $\hat{P}_2$ on the training set. Compute $T_i, i \in V$ by equation (4).
6. Perform sign test or two-sample t-test to test if $T < 0$.

---

## 3 NUMERICAL EXPERIMENTS

**Data Generation**  To evaluate the performance of SigClust-DEV, we conduct comprehensive simulation studies with observations generated from Bernoulli, Poisson, Poisson log-MVN, and Multinomial distributions. The parameter settings can be found in Appendix B. Notably, Poisson log-MVN and Multinomial distributions are commonly used to model the zero-inflated and over-dispersed scRNA data (Townes et al., 2019; Grabski et al., 2023). The null distribution is generated by one of the above distributions with fixed parameters, while the alternative distribution includes the mixture of distributions with different parameters. To investigate the impact of sample size, we vary the number of observations in $n \in \{100, 1000, 5000\}$, with the number of variables fixed at $p = 1000$.

**Evaluation Metrics**  We compare the empirical distribution of $p$-values with the uniform distribution. Under the null distribution where the data are unimodal, the empirical distribution of $p$-values is expected to be close to the uniform distribution on $[0, 1]$. Conversely, if the data are multimodal, the empirical distribution of $p$-values should be close to 0. To show the advantage of SigClust-DEV, we include the existing SigClust using hard-thresholding and soft-thresholding (SigClust-Hard and SigClust-Soft), MDS-based SigClust (SigClust-MDS), and scSHC for comparison.

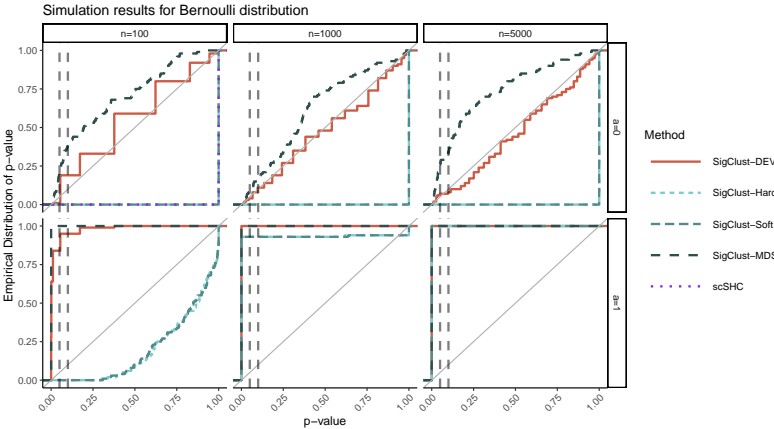

Figure 1: Empirical distributions of $p$-values from SigClust methods under simulation across 100 repetitions. In each panel, a mixture of two Bernoulli distributions is generated, where $a$ represents the variation between the two distributions (e.g., $a = 0$ indicates no cluster structure). SigClust-DEV performs best under both null and alternative hypotheses. The scSHC fails to produce $p$-values under the binary settings due to issues related to covariance estimation.

**Results for Simulation** As shown in the empirical distributions of $p$-values in Figure 1, SigClust-MDS exhibits inflated Type-I error rates under the null hypothesis across all sample settings, while scSHC is not applicable to binary data. In contrast, SigClust-Soft, SigClust-Hard, and SigClust-DEV effectively control the Type-I error rate under the null. Notably, the empirical $p$-value distribution of SigClust-DEV aligns closely with the diagonal, while the other methods tend to produce conservative $p$-values. Under the alternative hypothesis, the $p$-value distributions of SigClust-MDS and SigClust-DEV shift rapidly toward the upper-left corner, demonstrating greater statistical power compared to other methods, particularly when the sample size is small ($n = 100$). Additionally, SigClust-Soft and SigClust-Hard consistently fail to reject the null hypothesis, even when the data exhibit a clear cluster structure. The results for Poisson distribution are similar and are left in Appendix B.

Apart from its strong performance in the aforementioned distributions, SigClust-DEV is more powerful when the data distribution deviates from the exponential family, as seen in genomic datasets. Following Grabski et al. (2023), we assess the performance of SigClust-DEV for cell-type annotation using scRNA data modeled by Poisson log-MVN models. As shown in Figure 2, classical SigClust methods, which assume the data follow a Gaussian mixture, fail to provide reliable results. Specifically, SigClust-MDS does not maintain the correct statistical size under the null hypothesis, while SigClust-Soft and SigClust-Hard fail to reject the null under the alternative hypothesis. In contrast, SigClust-DEV and scSHC effectively detect false clusters, reducing the risk of over-clustering. However, scSHC exhibits limited power under the alternative hypothesis, becoming overly conservative when the sample size is small ($n = 100$) or the differences between clusters are modest ($a = 0.4$). Compared to SigClust-DEV, scSHC may miss novel cell types in biomedical research.

The multinomial distribution has also been used to model scRNA data from multiple libraries (Townes et al., 2019). Accordingly, we simulate unique molecular identifiers (UMIs) for one or two cell types across two libraries using a multinomial distribution. Figure 3 illustrates the performance of different SigClust methods in this scenario. Classical SigClust methods are misled by library effects, leading to false identification of new cell types when only one true cell type is present. Although scSHC is designed for multi-batch scRNA counts, it struggles to differentiate two cell types, sometimes missing novel cell types. This limited power highlights the constraints of scSHC's strong parametric assumptions. In contrast, SigClust-DEV effectively mitigates over-clustering while retaining its ability to discover new populations.

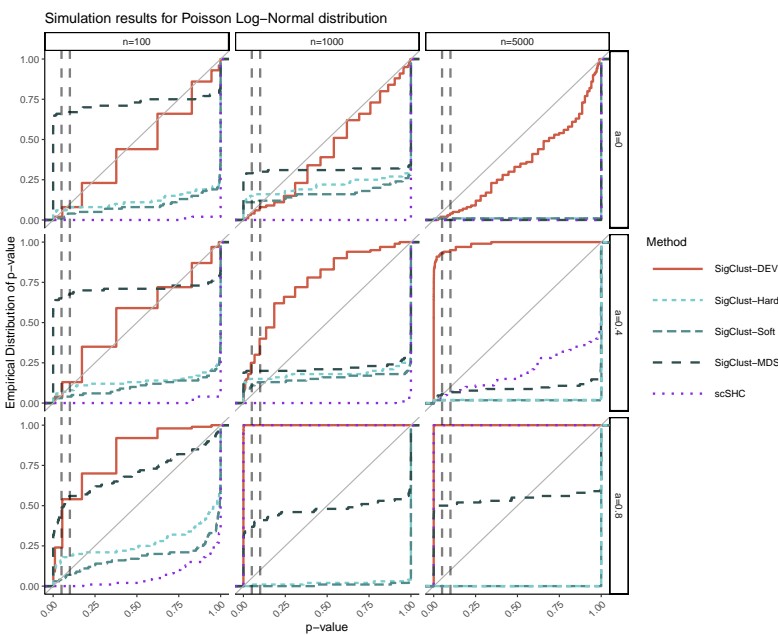

Figure 2: Empirical distributions of $p$-values from SigClust methods under simulation across 100 repetitions. In each panel, a mixture of two Poisson log-MVN distributions is generated, where $a$ represents the variation between the two distributions (e.g., $a = 0$ indicates no cluster structure). SigClust-DEV performs best under both null and alternative hypotheses.

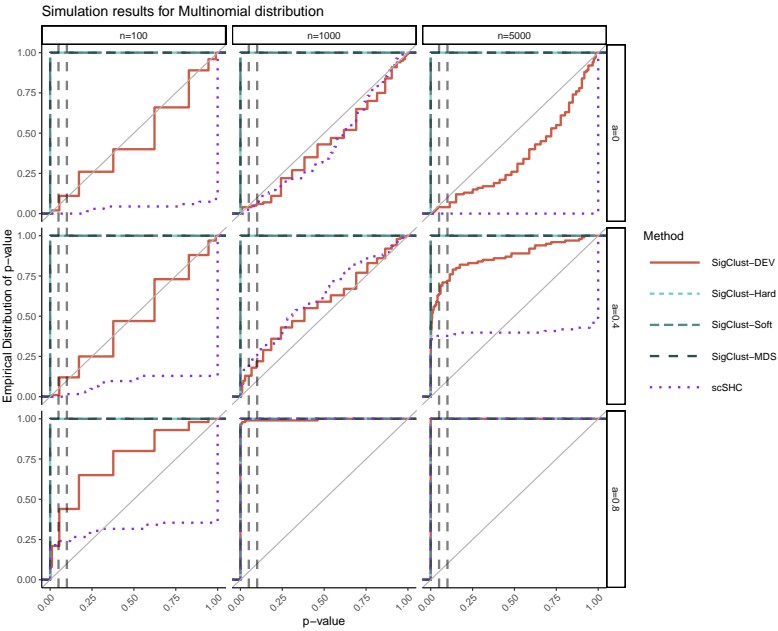

Figure 3: Empirical distributions of $p$-values from SigClust methods under simulation across 100 repetitions. In each panel, a mixture of two Multinomial distributions from two libraries is generated, where $a$ represents the variation between the two distributions (e.g., $a = 0$ indicates no cluster structure). SigClust-DEV performs best under both null and alternative hypotheses.

# 4 REAL DATA APPLICATIONS

In this section, we apply our proposed SigClust-DEV to two real datasets: *Hydra* scRNA data in Section 4.1 and EHR data in Section 4.2.

## 4.1 ANALYZING STEM CELL DIFFERENTIATION IN HYDRA

**Hydra Data Description**    The cnidarian polyp *Hydra* continuously self-renews and can regenerate its entire body from a small fragment of tissue. To investigate the molecular diversity of *Hydra* cells and the underlying transcriptional programs, Siebert et al. (2019) generated 24,985 *Hydra* single-cell transcriptomes from six libraries. Four libraries were generated using the original Drop-seq beads, while the other two libraries were generated using R&D beads. Specifically, *Hydra* consists of three cell lineages – endodermal epithelial, ectodermal epithelial, and interstitial – and each lineage is supported by its own stem cell population (Bosch et al., 2010). Epithelial stem cells further differentiate to build the foot (basal disk and peduncle) at the aboral end, body column, and the hypostome and tentacles at the oral end. Gene expression in these cells is constantly changing based on their positional context.

In this study, we investigate the differential gene expression of epithelial stem cells in Hydra with respect to their positions. Siebert et al. (2019) classified epithelial cells into six sub-populations: basal disk, peduncle, body column, hypostome, tentacles, and battery cells. However, their clustering results using Seurat (Butler et al., 2018) may not fully capture the biological distinctions between these populations. To address this, we need to answer the following questions:

1. Is the distribution of gene expression within a pre-identified cluster homogeneous?
2. Are the gene expression profiles of stem cells from the identified clusters significantly different from one another, and can any of these clusters be merged?

Specifically, we focus on the ectodermal cell lineage, which have two major libraries coded as 02 and 11 and five cell clusters of body column, peduncle, head/hypostome, battery cell, and basal disk. To evaluate the performance of SigClust-DEV and other comparison methods, we apply them for cells (i) from one single cluster, (ii) from two clusters. It is expected that SigClust produces large $p$-values for (i) and produces small $p$-values for (ii). Furthermore, since SigClust-Hard, SigClust-Soft, and SigClust-MDS do not account for the batch effect, for fair comparison, we also implemented these methods for cells in each library. Remarkably, body columns and battery cells are manually merged from multiple clusters, while the other clusters are from the output of clustering algorithm in Seurat (Butler et al., 2018).

**Results for Hydra Data**    Overall, SigClust-DEV and scSHC produce the most reasonable $p$-values. For the first question, we examine the $p$-values for cells from a single cell type (see Figure 4). SigClust-DEV and scSHC identify most of the annotated clusters coming from a unimodal count distribution except for battery cells, well-aligned with results in Siebert et al. (2019). On the other hand, although the original results merge two subclusters of battery cells, their gene expressions are evidently heterogeneous across the subclusters in Siebert et al. (2019)'s visualization using t-distributed stochastic neighbor embedding (t-SNE, Van der Maaten & Hinton (2008)). In contrast, SigClust-Soft, SigClust-MDS, and SigClust-Hard fail to preserve the size of testing and keeps rejecting the null hypothesis due to the batch effect. On the other hand, the significance of clustering seems to be inconsistent across libraries. For cells in library 11, SigClust-Soft and SigClust-Hard identify body column, peduncle, and basal disk as a single cluster, while suggesting battery cells and hypostome exist meaningful subgroups. However. for cells in library 2, they keep rejecting the null hypotheses for all cell types.

For the second question, we examine the clustering significance $p$-values for cells from multiple cell types (see Figure 4). The results of SigClust-DEV show that all mixtures of cell types are significantly separated, which well aligns with the manual annotations. In contrast, scSHC merges three cell types into one: body column, peduncle, and hypostome. Although we highlight that the biological groundtruth may be slightly different from the annotations, it is important to note that the molecular difference between *Hydra*'s body column and head has been widely observed for a long time (Holstein et al., 1991). Therefore, the results from scSHC may be an artifact of its power

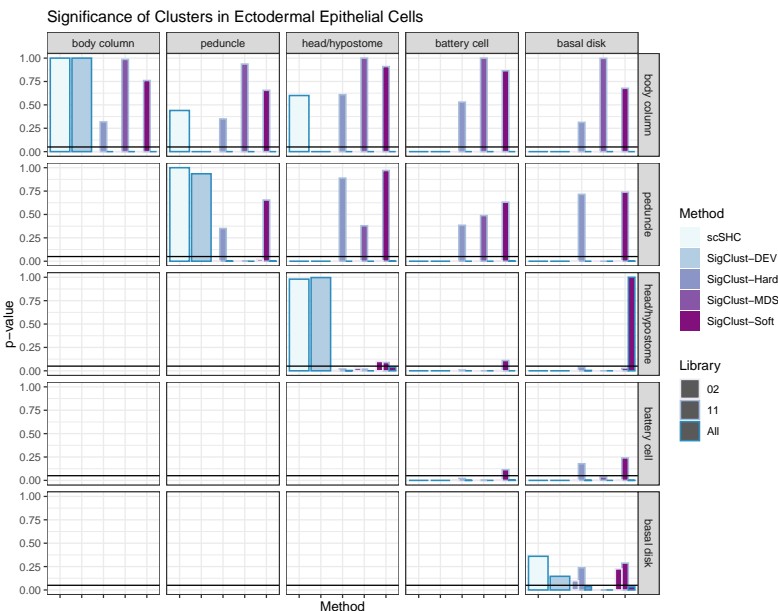

Figure 4: Clustering significance $p$-values for ectodermal epithelial cells in *Hydra*. The diagonal panels present the significance of one cell type, and the upper-right panels present the significance of multiple cell types. Since SigClust-Hard, SigClust-Soft, and SigClust-MDS do not account for the batch effect, for fair comparison, we also implemented these methods for cells in each library. SigClust-DEV and scSHC preserve the size under null distributions in most cases, and SigClust-DEV is more powerful than scSHC under the alternative in two cases on the first row.

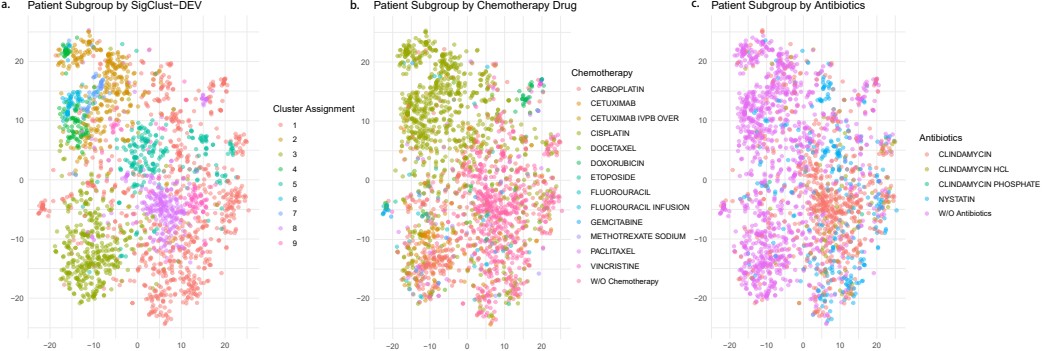

Figure 5: Treatment assignments for head-and-neck cancer patients using EHR. Each point represents the two-dimensional embedding of a patient's medical profile from t-SNE. (a) Clustering results using hierarchical clustering and SigClust-DEV on EHR data. (b) Patient subgroups stratified by chemotherapy drug usage. For patients who received multiple chemotherapy drugs, one was randomly assigned. (c) Patient subgroups stratified by antibiotic usage. For patients administered multiple antibiotics, one was randomly assigned.

issue. For other SigClust methods, we mainly focus on their results in library 11 due to their failure in statistical size in the other library. Interestingly, all these methods try to merge other cell types into the body column, including the battery cells, which contradicts with their previous findings that deny the homogeneity of their gene expression.

## 4.2 LEARNING LATENT MEDICAL GROUP STRUCTURE USING EHR

**EHR Data Description**   The dataset consists of medical histories from 7,284 head-and-neck cancer patients from a university hospital. After excluding patients with no records of using chemotherapy drugs or antibiotics, we are left with 2,203 patients and 973 different types of medications. This dataset, as a 2,203 by 973 matrix, allows for the comparison of the effectiveness of various treatments for head-and-neck cancer. However, the large number of medications presents a significant challenge. To address this, we propose applying hierarchical clustering and SigClust-DEV to this data matrix of medicine counts, to identify latent treatment patterns, i.e., subgroups of patients using similar medications.

**Results for EHR Data**   The combination of hierarchical clustering and SigClust-DEV reveals distinct patterns in drug usage among head-and-neck cancer patients (Figure 5a). Notably, it preserves patient subgroups stratified by the use of chemotherapy drugs and antibiotics (Figures 5b, c). Patients with less severe tumors, who do not undergo chemotherapy, are grouped into clusters 1, 8, and 9 in Figure 5a, highlighting the heterogeneity in drug usage among patients with benign tumors. Interestingly, the primary distinction between cluster 1 and clusters 8/9 appears to be the use of antibiotics, such as Clindamycin. In contrast, most patients undergoing chemotherapy are administered cisplatin, which corresponds to clusters 2, 4, 6, and 7. For patients that cisplatin is unsuitable, alternative treatments—such as cetuximab, carboplatin, and paclitaxel—are typically used in combination, and these correspond to cluster 3.

## 5 DISCUSSION

In this article, we propose a novel deviance-based SigClust method for testing the statistical significance of clustering for high-dimensional count data. Through learning the representation of the natural parameters of the count, our method avoids the estimation of the high-dimensional covariance matrix as in the original SigClust. Furthermore, to relax the requirement for Gaussian latent space, we test the relative goodness of fit between a single Gaussian distribution and Gaussian mixtures. This extension of SigClust makes it more broadly applicable in biomedical research.

There are several open questions for future research. Although we demonstrate the effectiveness of SigClust-DEV empirically, an interesting direction is to derive the theoretical conditions for the latent space properties. For instance, we observe that dimension reduction approaches like t-SNE can create spurious clusters by separating data from one single distribution, therefore their latent space is not suitable for SigClust. Another direction is to combine SigClust-DEV with hierarchical clustering to obtain more structured subgroup identification.

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

# A    METHOD DETAILS

**CI-Based SigClust on Generalized PCA Space**    To demonstrate the sensitivity of CI-based Sig-Clust to its Gaussian assumption, we also implemented Algorithm 2. The dimension reduction step is the same as SigClust-DEV, while the test statistic is replaced by $CI$.

---

**Algorithm 2** CI-based SigClust on the Generalized/Deviance PCA Space (SigClustCI-DEV)

---

1. Set the dimension of generalized/deviance PCA space $s$ and the dimension of MDS space $t$.
2. Obtain the latent variables $\mathbf{Z} = [\mathbf{z}_1, ..., \mathbf{z}_s]$ by solving (2) or (3).
3. Obtain the MDS matrix $\mathbf{Y} = [\mathbf{y}_1, ..., \mathbf{y}_t]$ from the dissimilarity matrix $\mathbf{D}$ of $\mathbf{Z}$ by solving (1).
4. Implement the $k$-means clustering on $\mathbf{Y}$ and compute the $CI$.
5. Estimate the sample covariance $\hat{\mathbf{\Sigma}}_Y$ of $\mathbf{Y}$. Generate samples from $\mathcal{N}(\mathbf{0}, \hat{\mathbf{\Sigma}}_Y)$.
6. Perform step 2-5 on the generated samples.
7. Repeat step 5 and 6 for $N_{sim}$ times. Obtain the empirical distribution of $CI$s.
8. Compute the p-value using the empirical distribution.

---

**Connection between CI and Relative Goodness of Fit**    In this section, we show that $CI$ is approximately equivalent to a special test of relative goodness of fit with $H_0 : P \sim P_0$, $H_1 : P \sim \frac{1}{k}\sum_{a=1}^{k} P_a$, where $P_a \sim \mathcal{N}(\boldsymbol{\mu}_0, \sigma^2 \mathbf{I}_{p \times p})$ with $\sigma^2$ known and small enough. Notice that $CI$ is equivalent to the ward linkage, defined as:

$$CI_W = \frac{1}{n}\sum_{i=1}^{n}\left(\|\mathbf{x}_i - \bar{\mathbf{x}}\|_2^2 - \min_{a=1,...,k}\|\mathbf{x}_i - \bar{\mathbf{x}}^{(a)}\|_2^2\right) = \frac{1}{n}\sum_{i=1}^{n} A(\mathbf{x}_i) - B(\mathbf{x}_i).$$

We first show the relationship between $A(\mathbf{x}_i)$ and $\log \hat{P}_0(x_i)$, i.e., the fit under the null hypothesis.

$$A(\mathbf{x}_i) = -2\sigma^2 \log\left(\frac{1}{(2\pi\sigma^2)^{\frac{p}{2}}}\exp\left\{-\frac{1}{2\sigma^2}\|\mathbf{x}_i - \bar{\mathbf{x}}\|_2^2\right\}\right) - p\sigma^2 \log(2\pi\sigma^2)$$

$$= -2\sigma^2 \log \hat{P}_0(x_i) - p\sigma^2 \log(2\pi\sigma^2).$$

Next we show the relationship between $B(\mathbf{x}_i)$ and the log-likelihood under the alternative hypothesis, $\log\left(\frac{1}{k}\sum_{a=1}^{k}\hat{P}_a(x_i)\right)$. Since the minimization function can be approximated by the log-sum-exponential function, we have

$$B(\mathbf{x}_i) = -2\sigma^2 \log\left(\sum_{a=1}^{k} \exp\left\{-\frac{1}{2\sigma^2}\|\mathbf{x}_i - \bar{\mathbf{x}}^{(a)}\|_2^2\right\}\right) + O(2\sigma^2 \log k)$$

$$= -2\sigma^2 \log\left(\frac{1}{(2\pi\sigma^2)^{\frac{p}{2}} k}\sum_{a=1}^{k} \exp\left\{-\frac{1}{2\sigma^2}\|\mathbf{x}_i - \bar{\mathbf{x}}^{(a)}\|_2^2\right\}\right) - p\sigma^2 \log(2\pi\sigma^2) + O(2\sigma^2 \log k)$$

$$= -2\sigma^2 \log\left(\sum_{a=1}^{k} \frac{1}{k}\hat{P}_a(\mathbf{x}_i)\right) - p\sigma^2 \log(2\pi\sigma^2) + O(2\sigma^2 \log k).$$

By combining the results, the ward linkage can be expressed as

$$CI_W = \frac{1}{n}\sum_{i=1}^{n} 2\sigma^2 \left[\log\left(\sum_{a=1}^{k}\frac{1}{k}\hat{P}_a(\mathbf{x}_i)\right) - \log \hat{P}_0(x_i)\right] - p\sigma^2 \log(2\pi\sigma^2) + O(2\sigma^2 \log k)$$

$$= 2\sigma^2 \bar{T} - p\sigma^2 \log(2\pi\sigma^2) + O(2\sigma^2 \log k).$$

For any $\epsilon > 0$, by taking $\sigma^2$ to small enough, we have $CI_W \leq 2\sigma^2\bar{T} + \epsilon$. Therefore, $CI_W$ is close to $\bar{T}$ given the above mentioned model assumption. However, when $\sigma^2$ is assumed to be too small as in $CI_W$, the Gaussian mixture in $H_1$ always provides a better fit to the data. Therefore, the expectation of $CI_W$ under the null is larger than 0, and the corresponding null distribution should be estimated by the Monte Carlo approach in the original SigClust.

**Setting of the Latent Space in SigClust-DEV**  Following Grabski et al. (2023), we utilize the top 30 principal components from generalized PCA for clustering throughout the paper. In line with Shen et al. (2024), we set the dimensionality of the MDS space to 2 for simulation purposes, while for real data analysis, we use a dimensionality of 10. In practice, the dimension of the generalized PCA space $s$ may affect clustering performance and can be adjusted to a larger value as needed. Similarly, the dimension of the MDS space $t$ may influence the covariance matrix estimation in SigClust-DEV, and it is recommended to keep $t$ relatively small for optimal results.

# B  ADDITIONAL DETAILS FOR NUMERICAL EXPERIMENTS

## B.1  DATA GENERATION MODEL

**Bernoulli Distribution**  Under the null hypothesis, the data were generated from a Bernoulli distribution Binomial$(1, \mathbf{p}_d)$, where $\mathbf{p}_d$ was sampled from $\mathcal{U}_d(0, 1)$. Under the alternative hypothesis, half of the observations are generated the same way as under the null hypothesis. For the remaining half of the observations, 10% of the elements in $\mathbf{p}_d$ were resampled from a different uniform distribution $\mathcal{U}_{100}(0, 1)$, introducing the cluster structure.

**Poisson Distribution**  Single multivariate Poisson distributions Poisson$(\boldsymbol{\lambda}_d)$ and mixtures of two Poisson distributions were generated for the null hypothesis and alternative hypothesis, respectively. Specifically, under the null distribution, $\boldsymbol{\lambda}_d$ was sampled from the exponential of $\mathcal{N}(0, \mathbf{I}_{d\times d})$ and fixed across samples. Under the null hypothesis, half of the observations are generated the same way as under the null hypothesis. For the remaining half of the observations, 10% of elements from $\boldsymbol{\lambda}_d$ was further multiplied by exponential of $\mathcal{N}(0, a\mathbf{I}_{100\times100})$, where $a$ was set to $\{0.4, 0.8\}$.

**Poisson Log-MVN Distribution**  Poisson Log-MVN distribution has been widely used to model counts of scRNA sequences. Similar to the scenario of Poisson distribution, we generated the single-cell counts (i) under the null hypothesis, i.e., the data followed a Poisson log-MVN$(\boldsymbol{\mu}_d, \sigma^2\mathbf{I}_{d\times d})$; (ii) under the alternative hypothesis, i.e., the data followed a mixture of Poisson log-MVN. For simplicity, we set $\boldsymbol{\mu}_d = \mathbf{0_d}$, and $\sigma^2 = 1$ under the null distribution, while the cluster structure under the alternative hypothesis was introduced by multiplying the 10% of the elements of $\boldsymbol{\mu}_d$ by the exponential of a normal distribution $\mathcal{N}(0, a\mathbf{I}_{100\times100})$, where $a$ was set to $\{0.4, 0.8\}$.

**Multinomial Distribution** Under the null hypothesis, to simulate the batch effect in scRNA data, we generated two multinomial distributions with equal sample sizes, denoted as $\text{Mult}(z_b, \boldsymbol{\pi}_d)$ for $b = 1, 2$, where $b$ represents the batch index. The size parameters were set to $z_1 = 1000$ and $z_2 = 2000$ for each batch, accounting for the batch effect. The probability vectors $\boldsymbol{\pi}_d$ were generated using the $sigmoid(\mathcal{N}(\mathbf{0}_d, 2\mathbf{I}_{d \times d}))$ distribution. Under the alternative hypothesis, we generated four multinomial distributions with equal sample sizes, forming two batches and two clusters. The batch effect remained the same as in the null hypothesis. To introduce the cluster effect, we altered 10% of the elements in $\boldsymbol{\pi}_d$, which were generated from $sigmoid(\mathcal{N}(0, a\mathbf{I}_{100 \times 100}))$ and varied between clusters.

## B.2 Additional Results

**Simulation Results for Poisson Distribution** Figure 6 presents the empirical distributions of $p$-values under the null and alternative setting. Under the null setting, an effective test is supposed to exhibit the empirical distributions of $p$-values close to the diagonal line. SigClust-DEV, SigClust-MDS, and scSHC performs best under all scenarios, while SigClust-Soft and SigClust-Hard present inferior power in three cases.

**Comparison between CI and Relative Goodness of Fit** Figure 7 presents the empirical distributions of $p$-values for CI-based SigClust-DEV in simulation. While CI-based SigClust-DEV successfully preserves the Type-I error in most settings, it is anti-conservative for Binary data when $n = 100, 500, 1000$. The results align with our expectation that CI can be more sensitive to the possibly non-Gaussian latent space.

**Comparison between Generalized PCA space and Deviance PCA space** Figure 8 presents the empirical distributions of $p$-values for SigClust on the generalized PCA space (SigClust-GLM) in simulation. The performance of SigClust-GLM is comparable with SigClust-DEV in most cases with respect to statistical size and power, except for Bernoulli and Multinomial distribution when $n = 100$. The inflated Type-I error in such cases may be a result of the algorithm instability of generalized PCA, as we notice that generalized PCA fails to converge under such cases.

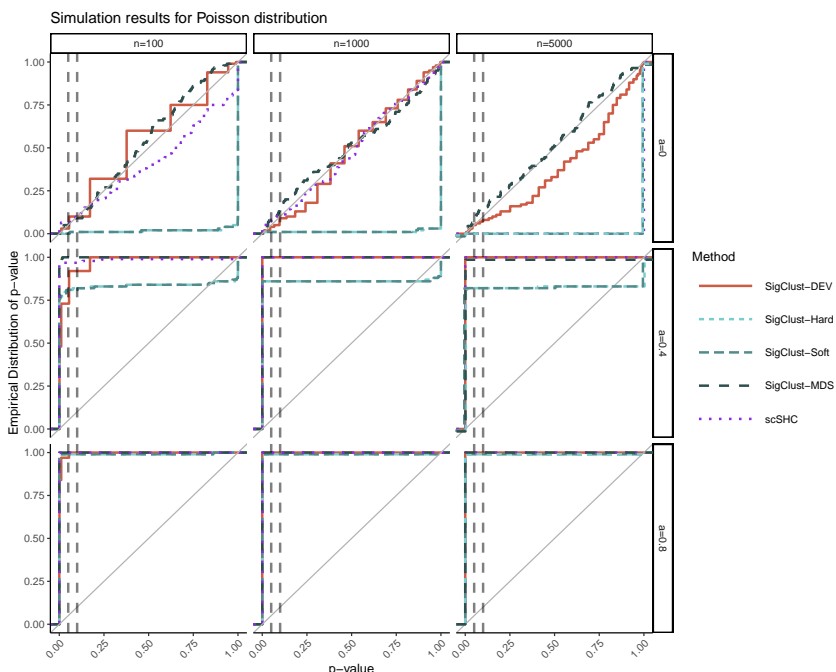

Figure 6: Empirical distribution of $p$-values from SigClust methods under simulation across 100 repetitions. In each panel, a mixture of two Poisson distributions was generated, where $a$ represents the variation between the two distributions (e.g., $a = 0$ indicates no cluster structure).

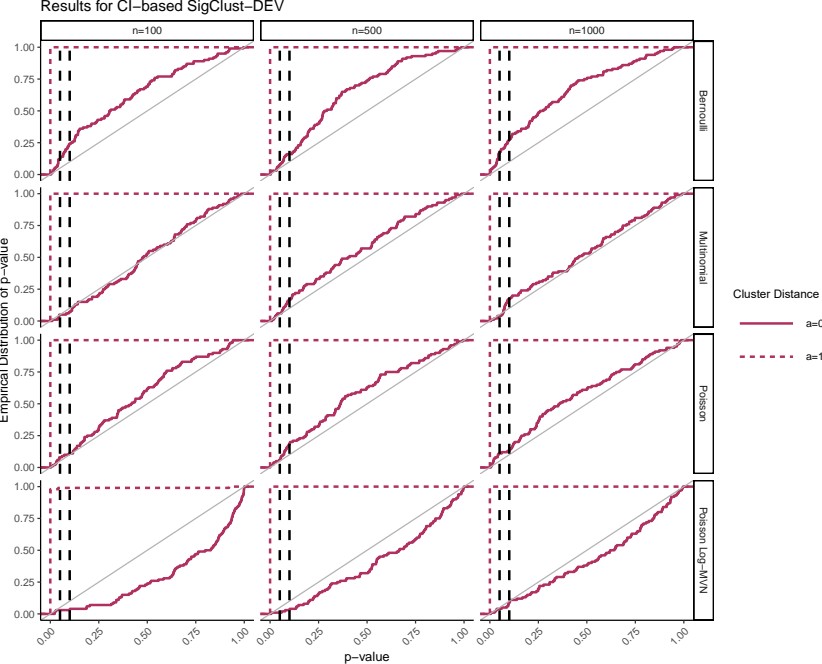

Figure 7: Empirical distribution of $p$-values from SigClust-DEV using CI as the test statistic across 100 repetitions. In each panel, a mixture of two distributions of its row was generated, where $a$ represents the variation between the two distributions (e.g., $a = 0$ indicates no cluster structure).

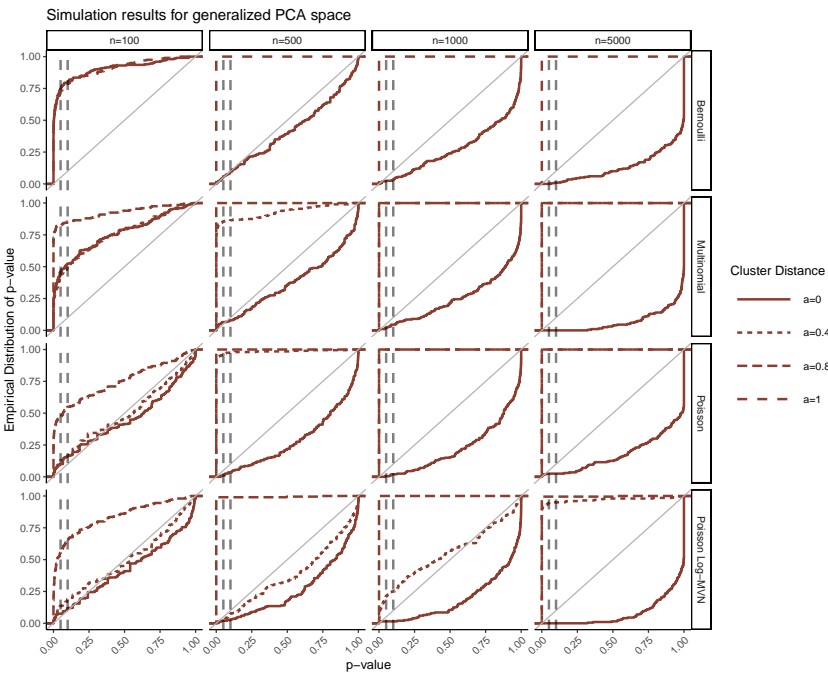

Figure 8: Empirical distribution of $p$-values from SigClust on generalized PCA space (SigClust-GLM) across 100 repetitions. In each panel, a mixture of two distributions of its row was generated, where $a$ represents the variation between the two distributions (e.g., $a = 0$ indicates no cluster structure). Note that generalized PCA can be unstable and fail to converge for small sample size such as $n = 100$.

