# OpenReview forum: "Statistical Significance of Clustering for High-Dimensional Count Data"
_ICLR.cc/2025/Conference — ICLR 2025 Conference Withdrawn Submission_

### Official Review · Reviewer_ZM4f · 2024-11-01

**Soundness:** 2
**Presentation:** 2
**Contribution:** 1
**Rating:** 3
**Confidence:** 3

**Summary:**

This paper proposes SigClust-DEV, a novel method for evaluating the statistical significance of clustering high-dimensional count data. It is tailored for applications in biomedical research where such data is prevalent. The proposed method uses DEV-PCA within a generalized PCA framework suited for count data from exponential family distributions.The method projects data into a low-dimensional latent space and assesses clustering significance by comparing the goodness of fit of single versus mixed Gaussian models on the projected data, avoiding direct covariance estimation challenges.

**Strengths:**

SigClust-DEV extends traditional SigClust methodology to handle high-dimensional count data, which is challenging due to over-dispersion and zero inflation in datasets like scRNA.

**Weaknesses:**

- A complexity and runtime analysis of the proposed method is missing.
- While comparisons to variants of SigClust and scSHC are presented, the paper doesn’t include comparisons to other recent clustering significance methods for count data.

**Questions:**

How does SigClust-DEV compare to other contemporary methods for testing clustering significance in count data (e.g., methods based on bootstrapping or permutation testing)?

---

### Official Review · Reviewer_ULio · 2024-11-01

**Soundness:** 2
**Presentation:** 3
**Contribution:** 3
**Rating:** 5
**Confidence:** 4

**Summary:**

This work combined the idea of generalized PCA (2002) that maximizes the likelihood of exponential family with a recent idea of MDS-based SigClust (2024).
Such a combination added flexibility in dealing with data types from exponential families than the past methods and is "more suitable" for the count data.

**Strengths:**

The motivation and idea of methodology are well presented.

**Weaknesses:**

1. Method and algorithm:
The main text needs more justification (results and discussion) to explain the choice of parameters ( i.e., the selected dimensions for PCA and MDS) and how they influence the algorithm's robustness. Consider providing sensitivity analyses showing how results change with different PCA/MDS dimensions.

2. Method and algorithm:
Computational complexity against past methods should be discussed (for scalability and feasibility in large datasets, especially in the modern data era). Consider providing run-time comparison against existing methods on datasets of varying sizes, or analyze the asymptotic complexity of SigClust-DEV compared to previous approaches.

3. Simulation:
Evaluation in noisy simulation settings should be added. Biological data, particularly scRNA-seq, and EHRs, often contain noise due to technical artifacts or measurement errors. It’s worth assessing how SigClust-DEV performs under varying noise levels and its robustness against noise. Some noises to consider may be dropout noises and Gaussian noises.

3. Real data:
More results on the EHR dataset regarding the performance of existing algorithms should also be added.

**Questions:**

How does the performance of other methods compare to the proposed method in EHR?
What impact do the parameters have on the robustness of the algorithm?
How does your method perform facing a mixture of Gaussian distributions (and, more generally, continuous data) compared to others? Is it specialized in count data?

---

### Official Review · Reviewer_wzXj · 2024-11-03

**Soundness:** 3
**Presentation:** 3
**Contribution:** 3
**Rating:** 5
**Confidence:** 4

**Summary:**

The paper presents a novel approach, SigClust-DEV, aimed at assessing the statistical significance of clustering in high-dimensional count data, with a focus on applications in biomedical research. The authors identify key limitations in existing clustering methods, specifically highlighting the inadequacy of traditional SigClust in handling discrete, non-Gaussian high-dimensional data such as genomic count data. By proposing a deviance-based extension of SigClust, the paper addresses these challenges and demonstrates the proposed method’s superiority through extensive simulations. Additionally, the application to real-world datasets, including scRNA sequencing and EHRs of cancer patients, adds practical relevance and robustness to the findings. Overall, this paper is well-organized and well-written.

**Strengths:**

The task of assessing the statistical significance of clustering in high-dimensional count data is of interest especially in the bioinformatic domain.
Writing and presentation skill is well.

**Weaknesses:**

SigClust-DEV first maps the data into a moderate-dimensional latent feature space Z using PCA, and then maps it further into another low-dimensional space Y using MDS. After two mappings, the data is no longer in a high-dimensional space, which may contradict the theme of the article?
The performance of this method and its practical application value should be evaluated on more diverse types of data.

**Questions:**

1. In the "1 INTRODUCTION" section, in the sentence "However, the significance of clustering for count data has not been thoroughly established," there seems to be a missing period before "However."
2. In the '2.1 CLUSTER SIGNIFICANCE FOR MIXTURE OF GAUSSIAN DISTRIBUTIONS' section, the formulas for CI and p-value are not numbered. Please ensure the numbering of formulas throughout the text is consistent.
3. Although this paper compares SigClust-DEV with existing SigClust variants, comparing it with other methods for assessing clustering significance for count data would provide a more comprehensive evaluation of the proposed method.
4. In section 2.3, SigClust-DEV first maps the data into a moderate-dimensional latent feature space Z using PCA, and then maps it further into another low-dimensional space Y using MDS. After two mappings, the data is no longer in a high-dimensional space, which may contradict the theme of the article? Additionally, in the subsequent implementation process, it would be helpful to supplement detailed parameter information for these two mappings.
5. In section 4.2 'LEARNING LATENT MEDICAL GROUP STRUCTURE USING HER', hierarchical clustering is chosen for exploration. Why not consider using other clustering methods?

---

### Note · Authors · 2024-11-13

I have read and agree with the venue's withdrawal policy on behalf of myself and my co-authors.